electrical engineering/mechanical engineering

aluminium conductor steel reinforced, sag calculation, forced convection, tensile stress distribution, creep effect, knee-point temperature

**Authors for correspondence:**
Liang Shi
e-mail: shi.l@gpdiwe.com
Gang Liu
e-mail: liugang@scut.edu.cn

[†]These authors were contributed equally to this work.

# Investigation of sag behaviour for aluminium conductor steel reinforced considering tensile stress distribution

Deming Guo[1,†], Pengyu Wang[1,†], Wencheng Zheng[1], Yang Li[2], Junwen Li[1], Wenhu Tang[1], Liang Shi[3] and Gang Liu[1]

[1]School of Electric Power Engineering, South China University of Technology, Guangzhou 510641, People's Republic of China
[2]State Grid Shijiazhuang Electric Power Supply Company, Shijiazhuang 050000, People's Republic of China
[3]Guangdong Hydropower Planning and Design Institute Co., Ltd, Guangzhou 510635, People's Republic of China

(iD) DG, 0000-0001-8703-1050

Sag calculation plays an important role in overhead line design. Since the tensile stress of aluminium conductor steel reinforced (ACSR) is required for the sag calculation, an analysis on sag behaviour when considering the tensile stress distribution can be very useful to improve the accuracy of sag results. First, this paper analyses the ACSR tensile stress distribution arising from the temperature maldistribution through proposing a new calculation formula. A finite-element analysis (FEA) model of ACSR is conducted for the solution of the new formula. By using the results, the error and limitations of the existing sag calculation methods for ACSR are discussed. As the critical point of sag calculation, knee-point temperature is solved iteratively involving the tensile stress maldistribution phenomenon in aluminium wires. Based on this iterative solution, an improved analytical method for the ACSR sag calculation considering the creep effect is presented and also compared with the hybrid sag method. The results show that these two methods are basically coincident without the consideration of creep effect, while there are non-negligible differences between them as the creep strain is involved. Compared with the existing analytical methods, the improved sag calculation method proposed in this paper can be applied in more extensive situations.

# 1. Introduction

The increasing electricity demand leads to the continuous growth of the power transfer capacity of overhead conductors. The overhead conductors face the risk of operating temperatures exceeding the thermal limit temporarily due to a deregulated environment [1]. The high operating temperatures of overhead conductors result in the increase of sag. When the sag exceeds the maximum allowable value, occasional blackout accidents occur in the overhead conductors [2–4]. Thus, to prevent excessive sag occurring in operating overhead conductors, it is essential to understand and accurately calculate the sag-temperature behaviour for reserving adequate safety margin in the overhead line design [5].

For composite conductors (e.g. aluminium conductor steel reinforced (ACSR)), the sag and the tension distribution are closely related. Due to the mismatch between the thermal properties of materials constituting the ACSR, the tension of aluminium wires gradually shifts to steel core with the increased conductor temperature [5]. Once the tensions of aluminium wires drop down to 0, the aluminium wires become slack. This phenomenon begins at a critical temperature called 'knee-point' temperature (KPT). For the case of conductor temperature being lower than KPT, all the tension is borne by the whole ACSR. While, for the case of conductor temperature being higher than KPT, all the tension is only borne by the steel core. Thus, the sag behaviour of ACSR is bilinear [6,7]. As a result, the accurate calculation of KPT is required to obtain sag-temperature behaviour.

Currently, various methods have been developed to calculate the sag-temperature behaviour of ACSR. Those methods are classified into two categories—the graphical methods and the analytical methods [5,7–10]. For the graphical methods [5,8,9], the tension of aluminium wires is specially introduced to estimate whether the conductor temperature reaches KPT, and it is usually calculated by the temperature and the stress–strain curves of conductor materials. Then, the bilinear sag behaviour of ACSR can be obtained. However, all aluminium wires are generally treated as a whole in the graphical methods. It means the temperature distribution in aluminium wires is ignored. A single tension (or tensile stress) is used for all aluminium wires in sag calculation. This is acceptable in the case of natural convection. However, the temperature distribution in aluminium wires under forced convection is distinct and non-negligible [8,11,12]. At present, there lacks focus on the calculation formula for the ACSR tensile stress distribution when the temperature maldistribution in aluminium wires is considered. Also, the KPT and sag calculation error caused by the temperature maldistribution in aluminium wires has not been discussed for the graphical methods.

The popular analytical methods include the numerical sag method [7] and hybrid sag method (HSM) [10]. Compared with the graphical methods, the tension of aluminium wires and stress–strain curves are not required for the analytical methods. In the numerical sag method, ACSR is modelled as a whole, and the mechanical and thermal properties of the overall ACSR are employed. However, the tension shifting between aluminium wires and steel core is without consideration. As a result, the numerical sag method shows a good accuracy in the case of conductor temperature below KPT, while a significant error is introduced into the method as conductor temperature exceeds KPT [7]. The numerical sag method is not competent to simulate the bilinear sag behaviour of ACSR.

HSM can accurately calculate the bilinear sag behaviour of ACSR under some assumptions [10]. For HSM, two parallel sets of iterations are carried out based on the mechanical and thermal properties of the overall ACSR and steel core, respectively. And the correct result is obtained by comparing with the results from the two sets of iterations. However, the creep effect is not taken into account in HSM. And the determination of KPT also requires a great deal of iterative computation. Thus, there still lacks a more applicative analytical method to simulate the bilinear sag behaviour of ACSR.

In this paper, the ACSR tensile stress distributions arising from the temperature maldistribution under different situations (including different current, creep strain and initial tension) are calculated and analysed by proposing a new formula. A finite-element analysis (FEA) model of ACSR is also established to obtain the temperature distribution. Subsequently, the errors and limitations of the existing sag calculation methods are detailed. Moreover, an iterative approach for calculating KPT is derived from the proposed calculation formula of tensile stress distribution. On the basis of the approach, an improved analytical method for calculating the bilinear sag behaviour of ACSR involving the creep effect is conducted, and the method is compared with HSM.

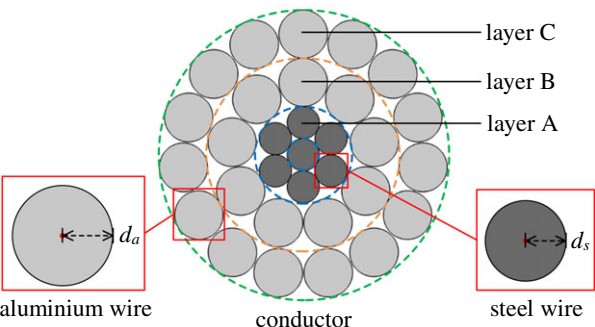

**Figure 1.** Radial geometry of 240/30 mm² ACSR.

**Table 1.** Structural parameters of 240/30 mm² ACSR.

| structural parameter | material |
| --- | --- |
| wire number of steel core | 7 |
| wire number of aluminium layers | 24 |
| $d_s$ (mm) | 1.20 |
| $d_a$ (mm) | 1.80 |

## 2. New formula for calculating aluminium conductor steel reinforced tensile stress distribution

Under the thermal effect of current, the overall temperature of ACSR rises. Nevertheless, the temperature rise of each wire is different, especially for the case of forced convection [8,12]. In addition, the operating conductor is subjected to an axial tension produced by self-weight. And a long-term creep also occurs in the conductor. As a result, the conductor can elongate along the conductor axis direction due to the effect of thermal expansion, tension and creep.

Since the coefficient of thermal expansion (CTE) and the elastic modulus of aluminium and steel are different, the free axial elongations of aluminium wires and steel core are different. Moreover, there still exists a free axial elongation difference between aluminium wires due to the variant temperature rise of each aluminium wire. However, constrained by the collaborative deformation of the whole conductor, all the wires show an equal axial elongation [6,13]. Consequently, the interaction forces between the wires appear. It causes the increased tensile stresses and axial elongations for wires with smaller free axial elongations, while the opposite case happens to the wires with larger free axial elongations [8].

Thus, to analyse the tensile stress in each wire of ACSR, a new formula considering the radial temperature distribution difference is proposed to calculate the ACSR tensile stress in the following. This paper takes a 240/30 mm² ACSR in figure 1 to specify the new formula, and the detailed structural parameters of 240/30 mm² ACSR are shown in table 1.

Two assumptions should be clarified for the derivation of the new formula as follows:

(1) elastic linear stress–strain behaviour is assumed for the steel and aluminium [5]; and
(2) the section area of each wire can be assumed as a constant since the radial thermal expansion and Poisson's ratio have little influence on its variation [14].

For any wire $n$ in figure 2, the relationship between the strains along the wire axis and the conductor axis is as follows [15]:

$$\varepsilon_{wn} = \varepsilon_{cn} \sin^2 \beta_n, \qquad (2.1)$$

where $\varepsilon_{wn}$ and $\varepsilon_{cn}$ are the strains along the wire axis and the conductor axis of wire $n$, respectively, and $\beta_n$ is the helix angle of wire $n$. It is noted that all the wires have the same $\varepsilon_{cn}$, referred to as $\varepsilon_c$ in the following

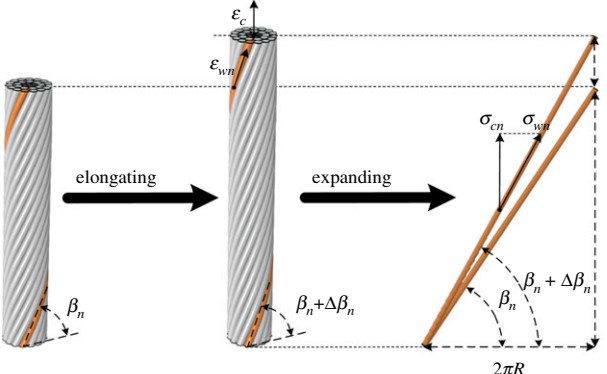

**Figure 2.** Elongations of ACSR and wire.

[6,13]. $\beta_n$ can be calculated by [15]

$$\beta_n = \arctan\left(\frac{l_n}{2\pi R_n}\right),\tag{2.2}$$

where $R_n$ is the shortest distance from wire $n$ to the conductor centre and $l_n$ is the pitch of wire $n$. $l_n$ can be obtained by the lay ratio, which is specified in IEC 61089 [16]. For the ACSR in figure 1, the selected lay ratios of layer A to layer C are 20, 15 and 14, respectively.

$\varepsilon_{wn}$ can be divided into three parts, including thermal strain, elastic strain caused by tension and creep strain [5,17]. Thus, the tensile strain of wire $n$ ($\varepsilon_{Fn}$) can be calculated by

$$\varepsilon_{Fn} = \varepsilon_{wn} - \alpha_n\Delta\theta_n - \varepsilon_n^{\text{creep}},\tag{2.3}$$

where $\alpha_n$ is the CTE for the material of wire $n$, $\Delta\theta_n$ is the temperature variation of wire $n$ with respect to the reference temperature for thermal expansion (i.e. 20°) and $\varepsilon_n^{\text{creep}}$ is the creep strain of material of wire $n$. According to [5], $\varepsilon_n^{\text{creep}}$ can be estimated by a thermal strain of a fixed equivalent temperature difference, as shown in (2.4),

$$\varepsilon_n^{\text{creep}} = \alpha_n\Delta\theta_{\text{creep}},\tag{2.4}$$

where $\Delta\theta_{\text{creep}}$ is the fixed equivalent temperature difference. This value is chosen based on utility field experience or calculations. For example, $\Delta\theta_{\text{creep}}$ is set to 15°C in Spanish utilities.

For wire $n$, the relationship between the tensile stress along the wire axis ($\sigma_{wn}$) and $\varepsilon_{Fn}$ is

$$\sigma_{wn} = \varepsilon_{Fn}E_n,\tag{2.5}$$

where $E_n$ is the elastic modulus for the material of wire $n$. The expanded wire $n$ along the direction of the helix angle is also shown in figure 2; then the relationship between the tensile stress along the conductor axis ($\sigma_{cn}$) and $\sigma_{wn}$ can be described as follows:

$$\sigma_{cn} = \sigma_{wn}\sin(\beta_n + \Delta\beta_n),\tag{2.6}$$

where $\Delta\beta_n$ is the increment of helix angle of wire $n$ due to the tension. The maximum value of $\Delta\beta_n$ can be estimated by the maximum allowable stress and elastic modulus of ACSR. The maximum allowable stress can be set as 40% of the rated tensile strength (RTS) [18], and the elastic modulus can be calculated by the elastic moduli of conductor materials and the structural parameters of ACSR [5,10]. Combined with (2.2) and (2.5), for any wire $n$, the maximum value of $\Delta\beta_n$ does not exceed 0.1°, while the value of $\beta_n$ has exceeded 80°. Since $\beta_n$ is far greater than $\Delta\beta_n$, (2.6) can be simplified to

$$\sigma_{cn} \approx \sigma_{wn}\sin\beta_n.\tag{2.7}$$

Combined with (2.1)–(2.3), (2.5) and (2.7), the calculation formula of $\sigma_{cn}$ considering the effect of radial temperature distribution and creep is obtained,

$$\sigma_{cn} = (\varepsilon_c\sin^2\beta_n - \alpha_n\Delta\theta_n - \varepsilon_n^{\text{creep}})E_n\sin\beta_n.\tag{2.8}$$

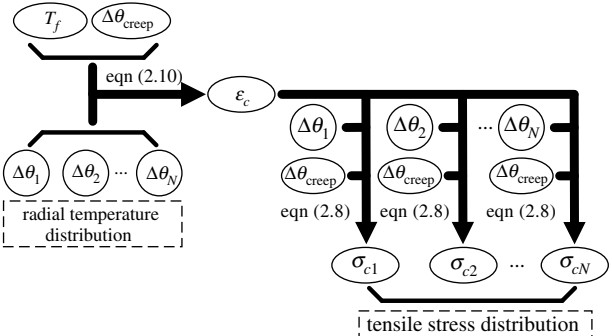

**Figure 3.** Solution procedure for calculating ACSR tensile stress distribution.

The initial tension of ACSR ($T_f$) is the sum of the tensions for all wires [17]. For wire $n$, its tension is equal to the product of $\sigma_{cn}$ and its section area $S_n$. $T_f$ can be also defined as some percentage $k$ of RTS [2,9,10], then

$$T_f = k \cdot RTS = \sum_{n=1}^{N} \sigma_{cn} S_n, \tag{2.9}$$

where $N$ is the wire number of the conductor. In (2.8), the value of $k$, which is limited by the maximum allowable stress condition, generally does not exceed 40% in practical application [7]. Substitute (2.8) into (2.9), then $\varepsilon_c$ can be conducted,

$$\varepsilon_c = \frac{T_f + \sum_{n=1}^{N} \alpha_n \Delta\theta_n E_n S_n \sin\beta_n + \sum_{n=1}^{N} \varepsilon_n^{\text{creep}} E_n S_n \sin\beta_n}{\sum_{n=1}^{N} E_n S_n \sin^3\beta_n}. \tag{2.10}$$

According to the new formula (i.e. (2.8)), the tensile stress distribution in ACSR considering the effect of temperature distribution can be calculated. The specific solution procedure for calculating ACSR tensile stress distribution is presented in figure 3.

It is indicated that the implementation of the new formula should depend on the known temperature variation of each wire. However, it is difficult to obtain the temperature variations of all the wires through experiments. Thus, FEA is adopted to achieve the solution.

# 3. Finite-element analysis calculation for aluminium conductor steel reinforced temperature

To obtain the steady-state radial temperature distribution of ACSR, a two-dimensional FEA model of ACSR based on the electromagnetic-thermal-fluid coupling is established in this section. Since the effect of radial temperature distribution difference of ACSR on the tensile stress calculation is focused in this paper, the case under the forced convection with obvious radial temperature distribution difference is investigated [8,9].

## 3.1. Modelling set-up

For the FEA model of ACSR shown in figure 4, the conductor is at the centre of circular air domain. To ensure the outer boundary temperature of circular air domain is consistent with the ambient temperature, the radius of circular air domain is set as 0.5 m [11].

The material parameters of ACSR are shown in table 2 [5,19]. For the value of steel relative permeability, the range is generally between 300 and 4000 due to different carbon content [20]. However, since the repartition of current between steel and aluminium is due to differences in material resistance, the steel relative permeability does not affect the current density distribution of ACSR. Thus, the steel relative permeability in table 2 is set to 1000.

For the FEA model of ACSR, the boundary conditions are set for thermal field and fluid field, respectively. The thermal field boundary conditions contain the heat radiation and heat convection on the surface of ACSR. The simulation of heat radiation is achieved by setting the aluminium emissivity $\varepsilon$ in the FEA model. For the new conductor investigated in this paper, $\varepsilon$ is set as 0.2 [21]. The simulation of heat convection is realized by the thermal-fluid coupling.

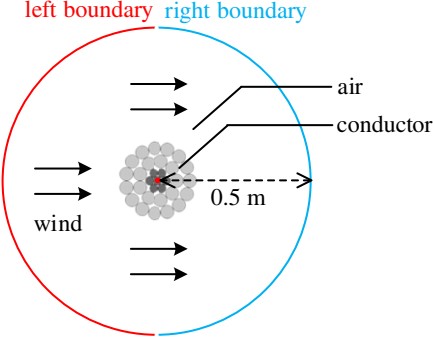

**Figure 4.** Structure schematic of FEA model.

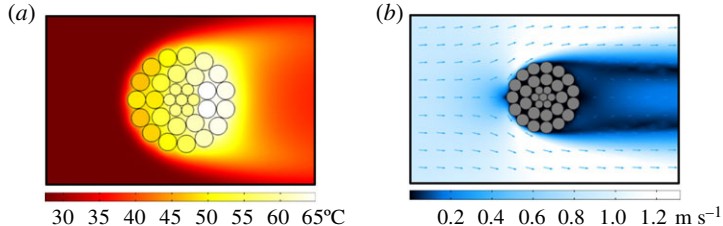

**Figure 5.** Calculation results of FEM: (a) temperature distribution and (b) wind speed distribution.

**Table 2.** Material parameters of ACSR (20°C).

| material | aluminium | steel |
| --- | --- | --- |
| conductivity (S m$^{-1}$) | $3.8 \times 10^7$ | $6.0 \times 10^6$ |
| relative permeability | 1 | 1000 |
| thermal conductivity (W m$^{-1}$ K$^{-1}$) | 238 | 44.5 |
| elastic modulus (GPa) | 69 | 196 |
| CTE (°C$^{-1}$) | $2.3 \times 10^{-5}$ | $1.2 \times 10^{-5}$ |

Black *et al.* [12] point out that for ACSR, the radial temperature difference mainly occurs in the aluminium layers, while the steel core (including the air gap inside the steel core) can be regarded as an isothermal body. Thus, apart from the air region in the steel core, the remaining air region of the FEA model is set as the fluid domain. As shown in figure 4, the left boundary of circular air domain is set as the air inlet, meanwhile, the right boundary is set as the air outlet. Then, the standard k-ε model of the turbulence model is adopted to solve the fluid field. The standard k-ε model can not only guarantee the accuracy of the calculation results, but also avoid a long computation time [22]. It is noted that the effect of air gravity is also considered during the solution of fluid field.

## 3.2. Modelling results

The radial temperature distribution result of ACSR under the current of 550 A, ambient temperature of 27.5°C and wind speed of 0.6 m s$^{-1}$ is obtained, as shown in figure 5a. Figure 5a indicates that the temperature distribution is axisymmetric along the horizontal direction. It means the effect of air gravity can be ignored. Figure 5a also points out that there exists an obvious temperature difference between aluminium wires from the same layer. It is caused by the uneven wind speed distribution around ACSR, as shown in figure 5b.

As the internal temperature difference in any wire is small, the temperature of wire centre is taken to analyse the temperature difference between different wires. To investigate the variation of temperature difference between aluminium wires from the same layer with current, the sampling wires in figure 6a are taken as the example, and the results under varying current for layer B and C are presented in

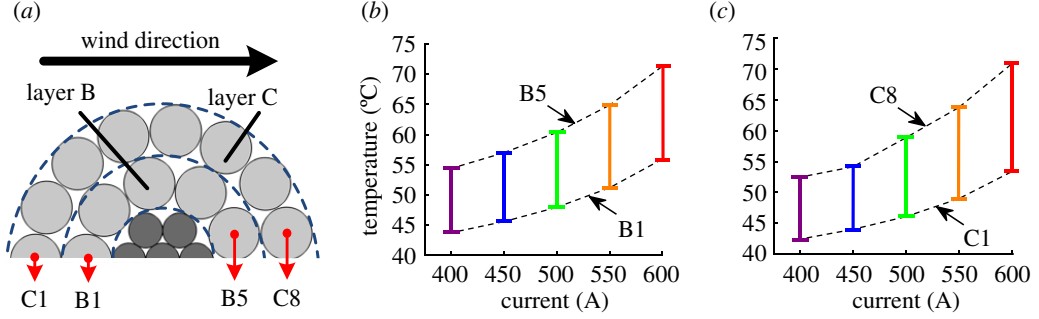

**Figure 6.** The temperature difference of the same aluminium layer under different current: (*a*) sampling diagram, (*b*) inner aluminium layer and (*c*) outer aluminium layer.

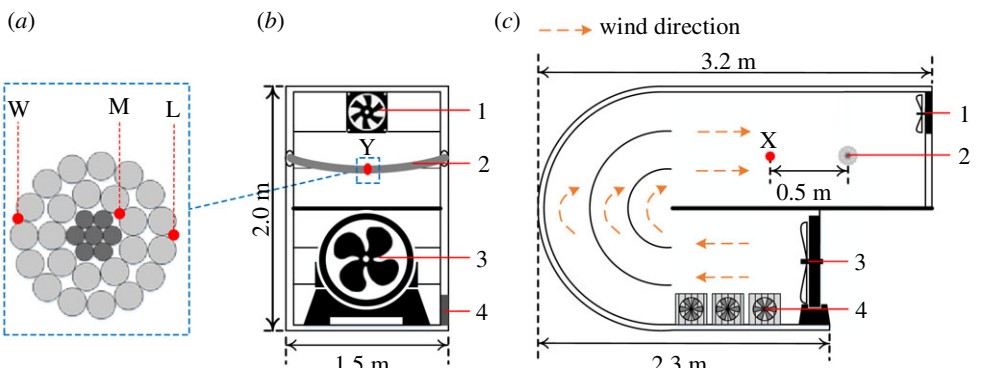

**Figure 7.** Structure diagram of wind tunnel experiment system (1 exhaust fan, 2 ACSR, 3 fan, 4 heat sink): (*a*) front view, (*b*) side view and (*c*) arrangement of temperature sampling points.

figure 6*b* and *c*, respectively. During the calculation of the FEA model, the wind speed is set for 0.6 m s$^{-1}$ constantly. For the aluminium wires from the same layer, the temperature variation with the current of the windward wires is larger than that of the leeward wires.

## 3.3. Modelling verification

The FEA model of ACSR is verified by the temperature rise experiments of a 15 m new conductor [23]. The conductor is formed into a closed loop by a parallel groove clamp, and part of it (length of 1.5 m) is set in a closed wind tunnel, as shown in figure 7*a*. For the wind tunnel, the wind is generated by a fan on the underside of wind tunnel, and the wind speed can reach up to 2 m s$^{-1}$. Moreover, the air in the tunnel is pumped out by an exhaust fan on the upside of wind tunnel to form a fluid circulation system. To monitor the wind speed near the conductor, an anemometer with the accuracy of 0.01 m s$^{-1}$ is placed 0.5 m away from the conductor (i.e. point X in figure 7*c*).

In the temperature rise experiments, the thermocouples with the accuracy of 0.1°C are used to measure the radial temperature distribution of ACSR. The arrangement of temperature sampling points is presented in figure 7*a*. To make the sampling results not being affected by the axial heat transfer, the three temperature sampling points in figure 7*a* are all located on the cross-section of the middle position of ACSR in the wind tunnel (i.e. point Y in figure 7*b*). In addition, to prevent the ambient temperature in the wind tunnel from rising due to the conductor heating effect, three heat sinks are placed near the fan to ensure the constant ambient temperature in the wind tunnel.

The temperature rise experiments are designed, and the detailed experiment conditions are shown in table 3. The duration of current for all temperature rise experiments is set to 70 min to guarantee the ACSR temperature can reach the steady state. Then, the steady state temperatures of sampling points under different conditions are measured.

The comparisons between the temperature results obtained from FEA model and experiments are presented in figure 8. It is noted that the values of current, wind speed and ambient temperature set in the FEA model are consistent with that measured in the experiments. Figure 8 shows that the calculation error of FEA model with respect to experiments under different conditions does not exceed

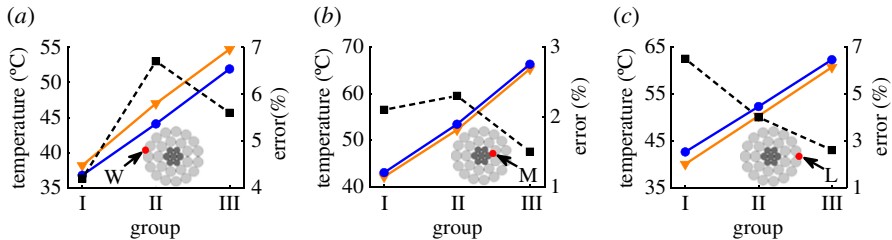

**Figure 8.** Comparison of simulation results and experimental results under different groups (the three curves corresponding to the experiment results (inverted triangle), simulation results (circle) and error (square), respectively): (*a*) point W, (*b*) point M and (*c*) point L.

**Table 3.** Conditions of temperature rise experiments.

| group | current (A) | wind speed (m s$^{-1}$) | ambient temperature (°C) |
| --- | --- | --- | --- |
| I | 350 | 0.5 | 25.5 |
| II | 450 | 0.5 | 26.3 |
| III | 650 | 1.0 | 27.5 |

7%, which indicates the FEA model has sufficient accuracy to simulate the radial temperature distribution of ACSR.

# 4. Results of new formula

## 4.1. Analysis for tensile stress distribution in aluminium conductor steel reinforced

For the wires from the same layer with constant creep strain, it is clarified by (2.8) that the tensile stress of wire is only the function of temperature. And figure 5*a* points out that the radial temperature distribution of ACSR is horizontally symmetric. Thus, only the tensile stress distribution for the upper half of ACSR needs to be analysed.

Based on the temperature results from the FEA model of ACSR shown in figure 5*a* and the assumption of $k = 20\%$ and $\Delta\theta_{\text{creep}} = 0$, the wire tensile stress results for different layers calculated by (2.8) are presented in figure 9. Figure 9 shows that due to the tiny temperature difference in layer A (i.e. steel core), the tensile stresses of wires from layer A are almost equal. However, for layer B or C, the tensile stresses of wires from the same layer are quite different, which are smaller than that from layer A. The uneven distribution of tensile stresses in aluminium layers is caused by the uneven wind speed distribution.

For the leeward aluminium wires with higher temperatures, their free axial elongations are larger than that of windward aluminium wires. However, all the wires present an equal axial elongation. It results that the leeward aluminium wires are subjected to the action of compression to reduce the axial elongations, while the windward aluminium wires are subjected to a greater tension to increase the axial elongations. Consequently, the tensile stresses of windward aluminium wires are greater than that of leeward aluminium wires, which is also indicated in figure 9.

## 4.2. Discussion

The temperature distribution, creep and $T_f$ of ACSR have a significant effect on the tensile stress distribution results. And figure 9 clarifies that the tensile stress distribution in the steel core is consistent, while the tensile stress maldistribution in aluminium layers is obvious. Thus, the tensile stress maldistribution phenomenon in aluminium layers under different currents, creep strains and $T_f$ are discussed, respectively, in the following.

Assume the wind speed is 0.6 m s$^{-1}$, the ambient temperature is 27.5°C, $k$ is 20% and $\Delta\theta_{\text{creep}}$ is 0, then the tensile stress distribution results of different aluminium layers under varying current are obtained, as shown in figure 10. Figure 10 indicates that for the leeward wires with higher temperature, the tensile stresses decrease with the increased current. It can be inferred that in the case of a large enough

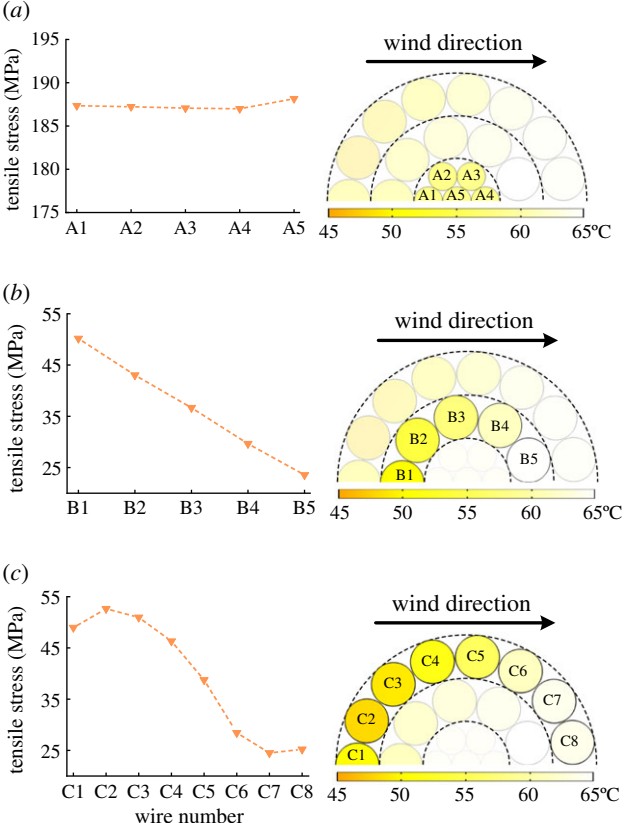

**Figure 9.** Tensile stress distribution in each layer of ACSR: (*a*) steel core, (*b*) inner aluminium layer and (*c*) outer aluminium layer.

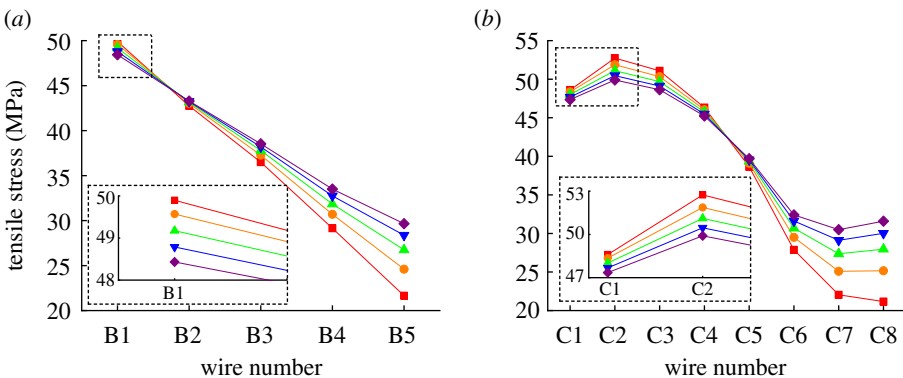

**Figure 10.** Tensile stress distribution of aluminium layers under different current (current is 600 A (square), 550 A (circle), 500 A (triangle), 450 A (inverted triangle) and 400 A (diamond), respectively): (*a*) inner aluminium layer and (*b*) outer aluminium layer.

current, the tensile stresses of leeward wires may drop down to 0. Also, the tensile stress maldistribution phenomenon in the same layer is intensified with a larger current.

Assume the wind speed is 0.6 m s$^{-1}$, the ambient temperature is 27.5°C, $k$ is 20% and the current is 600 A, then the tensile stress distribution results of different aluminium layers under varying creep strain are obtained, as shown in figure 11. Figure 11 indicates that the tensile stresses decrease as $\Delta\theta_{creep}$ increases. Since the creep strain is represented by a thermal strain of a fixed equivalent temperature difference, it can be inferred that the tensile stresses of aluminium wires are inversely proportional to $\Delta\theta_{creep}$ combined with (2.4), (2.8) and (2.10). For the aluminium wires from the same layer, the negative proportionality coefficient is uniform. Moreover, figure 11 also shows that the tensile stresses of leeward aluminium wires may vanish in the case of a large enough creep strain.

Assume the wind speed is 0.6 m s$^{-1}$, the ambient temperature is 27.5°C, the current is 600 A and $\Delta\theta_{creep}$ is 0, then the tensile stress distribution results of different aluminium layers under varying $T_f$

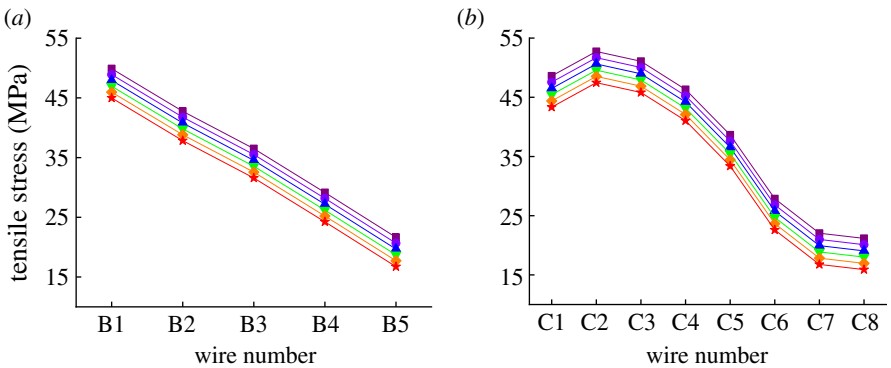

**Figure 11.** Tensile stress distribution of aluminium layers under different creep strains ($\Delta\theta_{creep}$ is 0℃ (square), 5℃ (circle), 10℃ (triangle), 15℃ (inverted triangle), 20℃ (diamond) and 25℃ (asterisk), respectively): (*a*) inner aluminium layer and (*b*) outer aluminium layer.

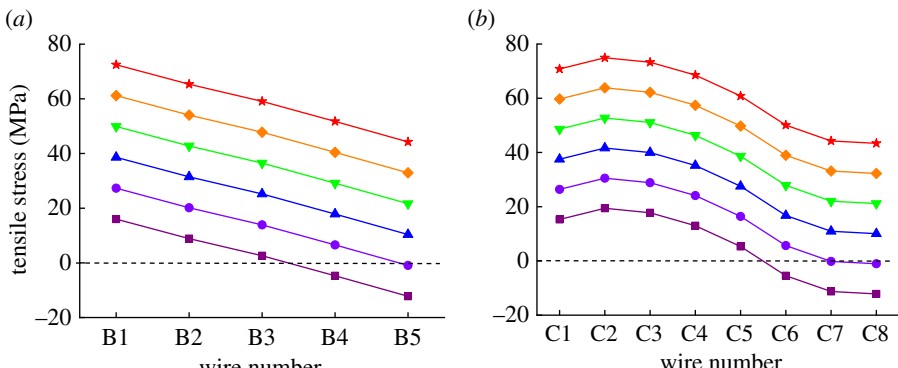

**Figure 12.** Tensile stress distribution of aluminium layers under different $T_f$ (*k* is 5% (square), 10% (circle), 15% (triangle, 20% (inverted triangle), 25% (diamond) and 30% (star), respectively): (*a*) inner aluminium layer and (*b*) outer aluminium layer.

are obtained, as shown in figure 12. Figure 12 indicates that the tensile stresses increase with the increased *k*, and the tensile stress variation of wires from the same layer is consistent. This phenomenon can be explained by combining with (2.8)–(2.10). From (2.8)–(2.10), it is inferred that the tensile stresses of all wires are proportional to the value of *k*, and the proportionality coefficient for wires from the same layer is uniform. Figure 12 also shows that in the case of a small enough *k*, the tensile stresses of leeward wires may disappear.

# 5. Analysis for existing sag calculation methods of aluminium conductor steel reinforced

The KPT, which is determined by the tensile stress distribution in ACSR, is crucial for the bilinear sag behaviour of conductor. Its determination process is different in the graphical methods and the analytical methods.

For the graphical methods in [5,8,9], the solution of KPT is realized through introducing the tension of aluminium wires. In these methods, the tension of aluminium wires is obtained by the product of the corresponding tensile stress and section area. According to (2.8), the tensile stress of aluminium wires is a function of temperature. While in the practical application of the graphical methods, the aluminium wires are usually treated as a whole, and the temperature distribution of aluminium wires is not considered. This means the input temperature of aluminium wires is a single value. Thus, the calculated tensile stress of aluminium wires is also a single value, which is inconsistent with the distribution results discussed in §4.

Since the input temperature of aluminium wires is not specified in the graphical methods, the calculated tension of aluminium wires is different when choosing the temperature of each aluminium

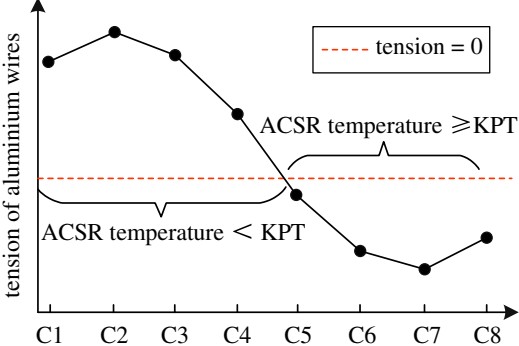

**Figure 13.** Tensions of aluminium wires under different wire temperatures from layer C.

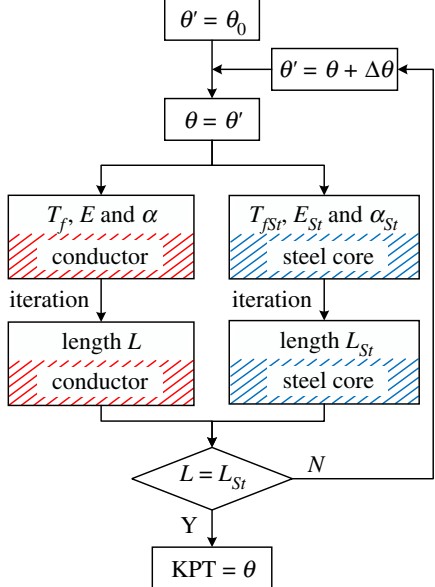

**Figure 14.** The procedure of HSM for KPT calculation.

wire as the input value. As a result, the error is inevitably introduced for the solution of KPT. It is indicated from §4 that in the case of a large enough current, creep strain or a small enough $T_f$, the tensile stresses of some leeward aluminium wires may disappear, while the windward aluminium wires are still subjected to tensile stresses, such as the case of $k = 5\%$ in figure 12. In this situation, assume the input conductor temperature is replaced by those of different wires from layer C, respectively, then based on the tension–temperature relation from graphical methods, the obtained tension characteristics of aluminium wires under different wire temperatures can be shown in figure 13. Figure 13 illustrates that the conductor temperature is considered to reach or even exceed KPT when choosing the temperatures of some leeward wires. The KPT is misjudged because some aluminium wires are still subjected to tension.

In addition, for the calculation of conductor sag under any temperature, the graphical methods depend on the stress–strain curves of conductor materials, although it takes many factors into account, such as the creep effect. This is also the main limitation of these methods.

For the analytical method in [10], i.e. HSM, the simulation of the bilinear sag behaviour for ACSR does not rely on the tension of aluminium wires. The KPT is determined by conducting two parallel sets of iterative calculations, as shown in figure 14. In the two calculation processes, the structural, mechanical and thermal properties (including section area, elastic modulus and CTE) of the whole ACSR and steel core are employed, respectively. The initial tensions of ACSR and steel core are also needed.

In figure 14, the initial conductor temperature is set to the reference temperature. Then, the length results of conductor and steel core from two sets of calculations are compared. The KPT is not

obtained until the results of two sets of calculations are equal. HSM shows a good accuracy under some assumptions. However, the creep effect is not considered for the calculation of KPT and conductor sag at any temperature. Since the solution of KPT starts at the reference temperature, a great deal of iterative computation is needed especially when KPT value is large.

# 6. An improved analytical method for sag calculation

Based on the analysis for the existing sag calculation methods of ACSR, an improved analytical method, which derives the calculation formula of KPT from (2.8), is proposed considering the creep effect and tensile stress maldistribution phenomenon in aluminium wires. And it also needs less iterations.

## 6.1. Solution of conductor 'knee-point' temperature

Any aluminium wire $n$ becomes slack when its tensile stress $\sigma_{cn}$ goes down to 0 [5]. Then, (6.1) can be inferred from (2.8),

$$\alpha_{Al}\Delta\theta_n + \varepsilon_{Al}^{\text{creep}} = \varepsilon_c \sin^2 \beta_n, \tag{6.1}$$

where $\alpha_{Al}$ and $\varepsilon_{Al}^{\text{creep}}$ are the CTE and creep strain of aluminium, respectively.

Due to the tensile stress maldistribution in aluminium wires, only when the aluminium wire with the largest tensile stress becomes slack, the whole ACSR can reach the 'knee-point' state [5,10]. It is indicated by figure 9 that the maximum of tensile stress in layer C is larger than that in layer B, which is also mentioned in [8,9]. Moreover, it is inferred from (2.8) that for the wire from the same layer, the tensile stress and temperature are inversely proportional. Thus, only in the case that the wire in layer C with the lowest temperature (hereinafter referred to as $\theta_{\text{low}}$) satisfies (6.1), the 'knee-point' state of ACSR is achieved. And the satisfactory $\theta_{\text{low}}$ can be defined as KPT, then (6.1) can be transferred to

$$\alpha_{Al} \cdot (\text{KPT} - 20) + \varepsilon_{Al}^{\text{creep}} = \varepsilon_c \sin^2 \beta_{\text{outer}}, \tag{6.2}$$

where $\beta_{\text{outer}}$ is the helix angle of the outer aluminium layer.

When the conductor temperature reaches KPT, steel core is subjected to all the tension. Thus, $\varepsilon_c$ can also be obtained by the elastic modulus, CTE and creep strain of steel core, as shown in (6.3).

$$\varepsilon_c = \frac{T_f}{S_{St} \cdot E_{St}} + \alpha_{St} \cdot (\text{KPT} - 20) + \varepsilon_{St}^{\text{creep}}, \tag{6.3}$$

where $S_{St}$, $E_{St}$, $\alpha_{St}$ and $\varepsilon_{Al}^{\text{creep}}$ are the section area, elastic modulus, CTE and creep strain of steel core, respectively.

Substitute (6.3) into (6.2), then the calculation formula of KPT can be obtained, as shown in (6.4). It is indicated by (6.4) that KPT is related to the initial conductor tension $T_f$ and the creep strain of conductor materials for a certain ACSR.

$$\text{KPT} = \frac{(T_f + S_{St}E_{St}\varepsilon_{St}^{\text{creep}})\sin^2\beta_{\text{outer}} - S_{St}E_{St}\varepsilon_{Al}^{\text{creep}}}{S_{St}E_{St}(\alpha_{Al} - \alpha_{St} \cdot \sin^2\beta_{\text{outer}})} + 20. \tag{6.4}$$

However, KPT also depends on the conductor length. This is because with the temperature increases, the conductor length increases and the conductor tension decreases. Thus, based on (6.4), an iterative process including conductor length is required to obtain the optimum KPT.

The optimum KPT calculation process is shown in figure 15. According to (6.4), the initial conductor KPT is calculated. As the conductor temperature reaches KPT, all the tension is borne by the steel core, and because of that, the total strain $\varepsilon_c$ can be calculated by (6.3). Then the conductor length $L$ can also be obtained by (6.5).

$$L = L_{\text{ref}}(1 + \varepsilon_c), \tag{6.5}$$

where $L_{\text{ref}}$ is the reference length. Its calculation process has been proposed in [10]. Based on the calculated $L$, the sag $D$ is obtained by (6.6), then a new $T_f$ can be calculated by (6.7) [10].

$$D = \sqrt{\frac{3s(L-s)}{8}} \tag{6.6}$$

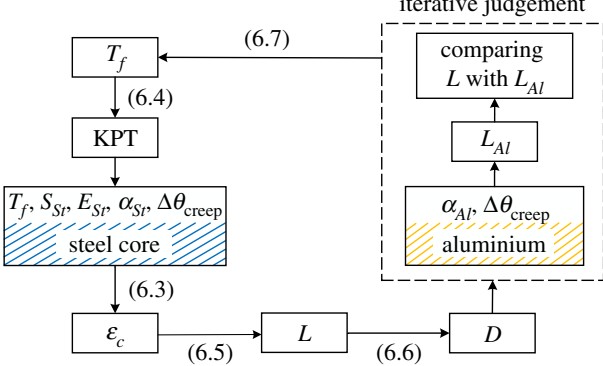

**Figure 15.** The procedure of the optimum KPT calculation.

and

$$T_f = \frac{ws^2}{8D},$$
(6.7)

where $w$ is the linear weight of ACSR and $s$ is the span length.

Since the thermal expansion is included, the new $T_f$ is lower in magnitude than the initial tension. The lower new $T_f$ reduces the value of the new KPT calculated by (6.4). This reduction in thermal elongation causes the sag to decrease and thus increases the corresponding $T_f$. An iterative process is introduced to obtain the optimum KPT, conductor tension $T_f$ and length $L$. In each iteration, the calculated conductor length $L$ is compared with the length of aluminium wires $L_{Al}$. KPT is iterated until the difference is below a threshold value (i.e. it is thought that $L$ is equal to $L_{Al}$), as shown in (6.8).

$$L_{Al} = [\alpha_{Al}(KPT - 20) + \varepsilon_{Al}^{creep} + 1]L_{ref} = L.$$
(6.8)

## 6.2. Sag calculation of the improved analytical method

For the case of $\theta_{low}$ below KPT, the tension is borne by the whole ACSR, while only the steel core is subjected to all tensions as $\theta_{low}$ exceeding KPT. This phenomenon can be mentioned as the temperature-tension behaviour of ACSR [10]. Combining the solution of KPT and the temperature-tension behaviour, an improved analytical method for sag calculation is proposed.

The flow chart of the improved analytical method is presented in figure 16, and the specific implementation procedures are as follows:

(1) Given the initial conductor tension $T_f$, the KPT is obtained by (6.4) and a set of iteration firstly. Then, taking the calculated KPT as the critical temperature, $\theta_{low}$ can be divided into two intervals. For each interval, different sag calculation procedures are adopted due to the temperature-tension behaviour of ACSR.
(2) If $\theta_{low}$ is below KPT, the calculation process of numerical sag method in [7] is applied to simulate the sag-temperature behaviour, as shown in figure 16. In this case, since the tension is borne by the whole ACSR, the structural, mechanical and thermal properties of ACSR (including section area, elastic modulus and CTE) are used during the sag calculations. Unlike the traditional numerical sag method, the calculation of $\varepsilon_c$ considers the creep effect, as shown in (6.9).

$$\varepsilon_c = \frac{T_f}{S \cdot E} + \alpha \cdot (KPT - 20) + \varepsilon^{creep},$$
(6.9)

where $S$, $E$, $\alpha$ and $\varepsilon^{creep}$ are section area, elastic modulus, CTE and creep strain of ACSR, respectively. For $E$ and $\alpha$, their formulae have been shown in [7]. $\varepsilon^{creep}$ is also estimated by an ACSR thermal strain of a fixed equivalent temperature difference.
(3) If $\theta_{low}$ exceeds KPT, the sag calculation procedure is similar to that in step 2. Since all tension is borne by the steel core in this case, the only distinction of procedure is that the structural, mechanical and thermal properties of steel core are used during the sag calculations. Similarly, the creep effect is also involved during $\varepsilon_c$ calculation, as shown in (6.3).

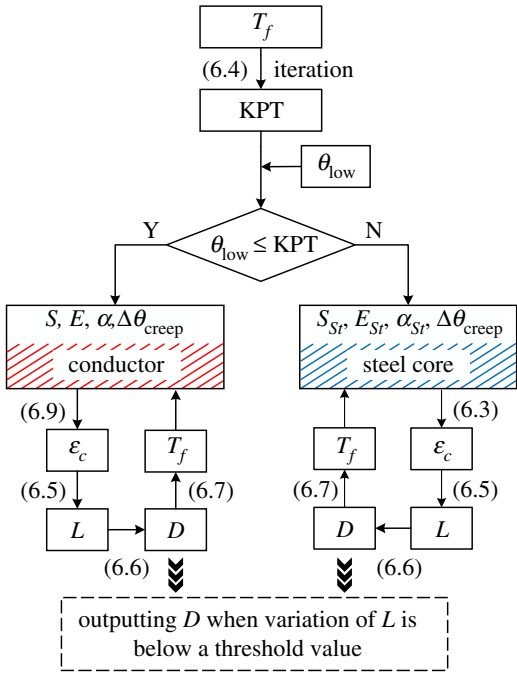

**Figure 16.** The procedure of improved analytical method for sag calculation.

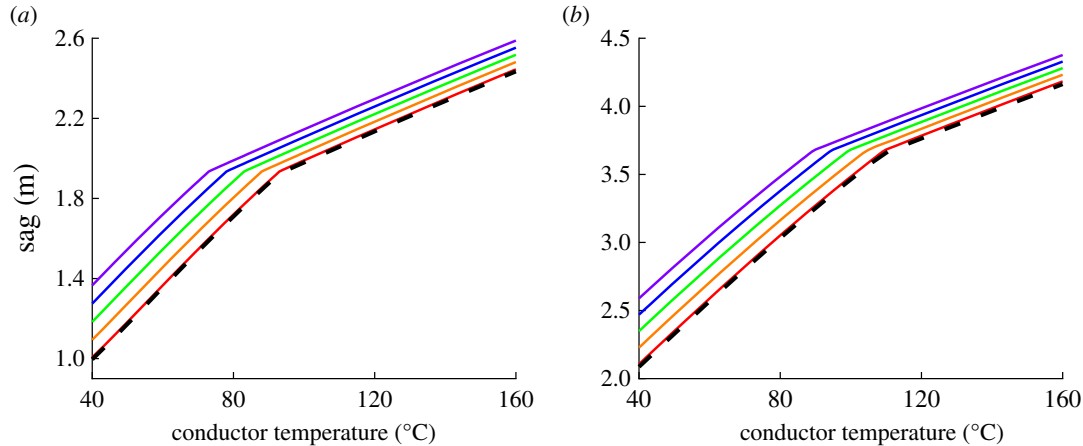

**Figure 17.** Comparison of sag calculation results between improved analytical method and HSM (full lines: improved analytical method with $\Delta\theta_{creep}$ of 0℃ (red), 5℃ (orange), 10℃ (green), 15℃ (blue) and 20℃ (violet); dotted lines: HSM): (a) span length of 100 m and (b) span length of 150 m.

Assume the $k$ is 20%, based on the proposed improved analytical method, the sag-temperature behaviours of ACSR under different span lengths and creep strains are obtained, as shown in figure 17. The results of HSM are also introduced for comparison (i.e. dotted lines in figure 17).

According to the calculation results of the improved method in figure 17, it is indicated that the sag increases with creep strain in the case of the same span length. And the KPT decreases as the creep strain becomes larger. This can be explained by figure 11. Since the tensile stresses of windward aluminium wires decrease with the increase of creep strain, the tension of aluminium wires goes down to 0 easily under a lower conductor temperature. Moreover, the KPT also increases with the span length.

In figure 17, in the case of $\Delta\theta_{creep} = 0$°C, the two result curves of the improved method and HSM are basically coincident. Consequently, the improved method without considering the creep effect shows a sufficient accuracy for HSM. While, the result curves of the improved method deviate from that of HSM when the creep effect is involved, and with the increase of creep strain, the results of the improved method deviate further. This also indicates that it is necessary to consider the creep effect in sag calculation.

Based on the above analysis, the advantages of proposed improved analytical method with respect to the existing sag calculation methods mentioned in §5 can be conducted. Compared with the existing sag calculation methods, the improved analytical method involves the tensile stress maldistribution phenomenon in aluminium wires when calculating KPT. Using this method, the bilinear sag behaviour of ACSR can be obtained without the stress–strain curves of conductor materials, and the creep effect is also taken into account.

# 7. Conclusion

This paper discussed the ACSR tensile stress distributions caused by the temperature maldistribution with different current, creep strain and initial tension by a proposed new formula. A coupled FEA model of ACSR was established to implement the formula, and it was verified by the temperature rise experiments. Based on the above analysis, the errors and limitations of implementing the existing sag calculation methods were discussed. Moreover, an iterative solution of KPT was derived from the proposed new calculation formula. Based on the iterative solution of KPT, an improved analytical method for calculating the bilinear sag behaviour of ACSR, which involves the creep effect, was presented. The results of the improved analytical method and HSM were also compared.

Data accessibility. Data (including the experimental and simulation data) and code are available as the electronic supplementary material. The parameters used in the simulation model and mathematical model are available in the papers cited.

Authors' contributions. D.G. was involved in investigation, validation and writing of original draft. P.W. was involved in conceptualization, formal analysis, supervision, editing and review of the draft. W.Z. was involved in simulation modelling and data curation. Y.L. was involved in software and formal analysis. J.L. was involved in simulation modelling and validation. W.T. was involved in project administration, review and editing. L.S. was involved in methodology, resources, editing and review of the draft. G.L. was involved in resources, funding acquisition, editing and review of the draft.

Competing interests. We declare that we have no competing interests.

Funding. This work was supported by the National Natural Science Foundation of China (grant no. 51977083) and the National High Technology Research and Development Program (863 Program) (grant no. 2015AA050201).

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
