## [Peer Review File · Royal Society Open Science]

Review History

RSOS-210049.R0 (Original submission)

Review form: Reviewer 1 (Gui Yun Tian)

Is the manuscript scientifically sound in its present form?

No

Are the interpretations and conclusions justified by the results?

No

Is the language acceptable?

Yes

Do you have any ethical concerns with this paper?

No

Have you any concerns about statistical analyses in this paper?

No

Recommendation?

Accept with minor revision (please list in comments)

Comments to the Author(s)

The paper is well structured. More explanation and experimental evaluation are required. More quantitative analysis could be provided.

Review form: Reviewer 2**Is the manuscript scientifically sound in its present form?**

Yes

Are the interpretations and conclusions justified by the results?

Yes

Is the language acceptable?

Yes

Do you have any ethical concerns with this paper?

No

Have you any concerns about statistical analyses in this paper?

No

Recommendation?

Accept as is

Comments to the Author(s)

The method described in the paper includes the effect of the wire stress variations with temperature for the sag-tension calculation. It is well described, it is relevant and it is useful.

Decision letter (RSOS-210049.R0)

Dear Dr Shi:

I am pleased to inform you that your manuscript entitled "Investigation of Sag Behavior for ACSR Considering Tensile Stress Distribution" is now accepted for publication in Royal Society Open Science.

on behalf of Dr Chong Li (Associate Editor) and Professor R. Kerry Rowe (Subject Editor).

Associate Editor Dr Chong Li Comments to Author:

Comments to the Author:

Dear Authors,

Thank you for submitting your work to Royal Society Open Science for consideration of publishing. Based on the feedback from our fellow reviewers, I would recommended the following minor changes for your manuscript.

1. delete lines 39-41 of page 3
2. Line 19 page 5: Define n
3. Correct temperature unit in brackets on y-axis of Fig. 6 b&c, Fig.8 (label of x-axis), and Fig.17
4. Correct naming order of the sub diagrams in Fig.7

Yours sincerely
Associate Editor
Dr. Chong Li

Reviewer(s)' Comments to Author:

Reviewer: 1

Comments to the Author(s)

The paper is well structured. More explanation and experimental evaluation are required. More quantitative analysis could be provided.

Reviewer: 2

Comments to the Author(s)

The method described in the paper includes the effect of the wire stress variations with temperature for the sag-tension calculation. It is well described, it is relevant and it is useful.
